# The Role of Simpson Grading System in Spinal Meningioma Surgery: Institutional Case Series, Systematic Review and Meta-Analysis

**DOI:** 10.3390/cancers17010034

**Published:** 2024-12-26

**Authors:** Giuseppe Corazzelli, Sergio Corvino, Valentina Cioffi, Ciro Mastantuoni, Maria Rosaria Scala, Salvatore Di Colandrea, Luigi Sigona, Antonio Bocchetti, Raffaele de Falco

**Affiliations:** 1Department of Neurosciences and Reproductive and Odontostomatological Sciences, Neurosurgical Clinic, University “Federico II” of Naples, 80138 Naples, Italy; sercorvino@gmail.com; 2Neurosurgery Department, Santa Maria delle Grazie Hospital, ASL Napoli 2 Nord, Via Domitiana Località La Schiana Pozzuoli, 80078 Naples, Italy; valecioffi81@gmail.com (V.C.); mastantuoniciro@gmail.com (C.M.); drscala.mariarosaria@gmail.com (M.R.S.); doct13@libero.it (L.S.); antonio.bocchetti74@gmail.com (A.B.); raffaele.defalco@aslnapoli2nord.it (R.d.F.); 3Department of Anaesthesiology and Intensive Care Medicine, Santa Maria delle Grazie Hospital, ASL Napoli 2 Nord, Via Domitiana Località La Schiana Pozzuoli, 80078 Naples, Italy; salvatore.dicolandrea@aslnapoli2nord.it

**Keywords:** spinal meningioma, simpson grade, systematic review, meta-analysis, meningioma recurrence, intradural extramedullary tumor

## Abstract

Spinal meningiomas are among the most frequent primary intradural extramedullary tumors, predominantly affecting women and the thoracic spine. While generally benign, these tumors require surgical intervention to achieve gross total resection and prevent recurrence. The Simpson grading system, originally developed for intracranial meningiomas, remains a pivotal factor in predicting recurrence risk, though its applicability to spinal meningiomas has been debated. This study combines a mono-institutional case series with a systematic review and meta-analysis to evaluate the prognostic significance of Simpson grading in spinal meningioma surgery. By analyzing recurrence rates and surgical outcomes, this work aims to provide evidence-based insights into optimal management strategies for this challenging pathology.

## 1. Introduction

Meningiomas are generally benign, slow-growing tumors that arise from arachnoid cap cells, although some lesions may be atypical and malignant. Available records in the literature claim spinal meningiomas represent 1.2 to 12% of all meningiomas and 25 to 45% of all intradural spinal tumors [1,2,3]. These lesions are described as intradural extramedullary tumors and mainly occur at the thoracic level in women aged between 50 and 80 [3]. The standard of care is the safe and precise surgical resection of the lesion, with satisfactory functional recovery and preservation of spinal stability [4]. The primary goal of surgery is complete tumor excision, typically Simpson grades I or II, along with spinal cord decompression [4,5]. Recent studies suggest that Simpson II, the current gold standard in spinal meningioma surgery, is associated with a significantly higher rate of symptomatic recurrences requiring reoperation [6,7]. Furthermore, accredited studies report that a higher Simpson grade is an independent increased risk factor for recurrence of spinal meningioma [8]. However, solid evidence is lacking in the literature regarding the objective risk of recurrence of spinal meningiomas following higher Simpson resection grades [8,9]. Introduced in 1957 to estimate the recurrence risk of intracranial meningiomas based on the extent of resection [10], nowadays, the Simpson grading still remains one of the most relevant predicting factors of recurrence [2,11,12,13], although its validity has been recently questioned by several authors [14,15,16,17]. As the extent of resection is a modifiable risk factor in terms of surgical aggressivity, it is very important to verify its validity in order to tailor the strategy of treatment for each patient.

In this setting, our question was as follows: more than half a century after its introduction, considering the refinements of the surgical techniques and pre- and intraoperative tools, as well as the increasing role of the molecular markers and related targeted therapies, is Simpson grading still valid as predicting factor of recurrence for spinal meningiomas? Can we plan the decision-making process of management according to it? The present study aims to answer these questions. Therefore, we investigated the existing evidence in the literature of the last four decades on the relevance of the Simpson grading system for spinal meningiomas and the weighted risk of recurrence for each Simpson grade through a detailed and comprehensive literature meta-analysis and a retrospective analysis of a mono-institutional surgical series of spinal meningiomas.

## 2. Methods

### 2.1. Study Setting

The study was conducted as a systematic review and meta-analysis of the literature from 1980 to 2023, using strict inclusion criteria and solid statistical methodology. An institutional series of 74 patients was described and included in the study. The analyses studied the weighted risk of recurrence among patients who received Simpson grade III or more against Simpson I and II, as well as the risk of recurrence between resection grades I and II.

### 2.2. Definition of the Institutional Cohort

Medical record data of patients with histological diagnosis of spinal meningiomas and operated on at the “Santa Maria delle Grazie” hospital between May 2006 and January 2023 were retrospectively reviewed.

The inclusion criteria were patients aged 18 years old or older who were operated on for spinal meningioma at first diagnosis, a reliable surgical Simpson grading system (Table 1), pre-and 6-month postoperative contrasted-enhanced spinal MRI availability, and a follow-up of at least one year.

All patients underwent detailed standard neurological examination at admission and after surgery by a neurosurgeon and pre- and postoperative (3 months after surgery) contrast-enhanced spinal MRI. All surgeries were performed through a standard posterior midline approach with laminectomy extended to include the rostral and caudal limits of the tumor, and the dura was opened on the midline. For Simpson grade I, the tumor dural attachment was resected and the dural defect was reconstructed with an artificial dural substitute. For Simpson grade II, the tumor dural attachment was coagulated along the extension of the dural tail sign at the beginning of the procedure for the deafferentation of the tumor and at the end after using the ultrasonic aspirator. The Simpson grade was defined based on the intraoperative view.

The histology was defined according to the 2021 WHO classification of tumors of the Central Nervous System. Patients’ baseline characteristics are listed in Table 2.

### 2.3. Literature Search Strategy

A systematic review was performed from January 1980 to May 2023. This study was not registered on PROSPERO or other similar databases. However, the study adhered to the Preferred Reporting Items for Systematic Reviews and Meta-Analyses (PRISMA) guidelines to ensure methodological rigor and transparency.

PubMed, EMBASE, Cochrane Library, Web of Science, and Google Scholar electronic online databases were searched for studies describing the surgical outcomes of operated spinal meningiomas to identify articles regarding an association between the Simpson grade of resection and the recurrence of spinal meningiomas. Keywords included “Spinal meningioma”, “Adult spinal surgery”, “Intradural extramedullary tumor”, “Laminectomy”, “Simpson grade”, and “Extent of resection”. The synonyms of Medical Subject Headings (MESH) terms and Boolean operators “AND” or “OR” were also used to search.

The search strategy was unrestricted by study design or publication date but included only English-language keywords. Additional articles were identified via citation searching. The titles and abstracts of all records were screened by G.C. and S.C. Two researchers (G.C. and S.C.) evaluated the level of evidence of observational studies using the Melnyk and Fineout-Overholt system [18], and another Author (A.B.) further resolved any conflict in data analysis and studies enrollment. The search strategy is listed in Table 3.

### 2.4. Selection Criteria

Two independent researchers (G.C. and S.C.) screened the literature according to titles and abstracts. After excluding irrelevant studies, the remaining abstracts were read for inclusion. Subsequently, a selection of articles was made, for which odds ratio (OR) calculations and prevalence analysis could be carried out based on the following inclusion and exclusion criteria: histological diagnosis of Spinal Meningioma, operated patients, unambiguous data related to patients’ Simpson resection grade for spinal meningiomas (Table 1), lesion’s recurrences after at least six months from the surgery, and one year stated follow-up minimum. Inclusion criteria for the type of studies were as follows: case series (CS) reporting at least twenty patients, retrospective (RCoh) or prospective cohort studies (PCoh), and case–control (CC) studies; reviews, systematic reviews, and meta-analyses, not describing a proper case series, were investigated for references.

Exclusion criteria included articles not clearly stating a follow-up time or the follow-up time being <1 year, patients lost at follow-up >20%, unknown or inaccurate data, multiple reports or repeated literature on the same population (in case of published data, we only included the most significant and conspicuous population), no data on control or relapse-related influencing factors, not reported Simpson resection grade, low-quality studies, the level of evidence being <6.

The majority of spinal meningiomas were classified as WHO grade I, reflecting their benign nature. Atypical (grade II) and malignant (grade III) lesions were rare and significantly impacted recurrence risk.

In the institutional cohort, tumor location and extension were recorded and considered in the analysis. However, due to inconsistent reporting across the reviewed studies, these factors could not be universally included in the meta-analysis.

While some studies included patients treated before 1980, all were published after 1980 and adhered to the predefined inclusion criteria, ensuring methodological consistency.

Recurrence was defined as tumor relapse for Simpson grades I and II, and lesion volume doubled for Simpson III, IV, and V, requiring reoperation at least six months after the first surgery. Therefore, those patients were the scope of our study. Table 4 lists the criteria and definitions used in the current meta-analysis to group complications reported among the eligible studies.

### 2.5. Data Extraction

Two researchers (G.C. and S.C.) independently extracted relevant data from eligible studies. The extracted variables included essential characteristics of the studies (Author, publication year, country of study, follow-up time, sample size, sex distribution), number of patients, number of recurrence-free patients, first surgery Simpson grade, age at surgery, level involved, and histological diagnosis. The data were incorporated into Excel (Microsoft Corporation, Redmond, WA, USA; Version 2016).

### 2.6. Objectives

The study aimed to investigate the prognostic value of the Simpson grading for spinal meningiomas and the acceptableness of Simpson II resection for spinal meningiomas. Studies enrolled in the different statistical tests are described in Table 5.

### 2.7. Statistical Analysis

After selecting articles suitable for the analysis, articles were classified into two groups. The first group included the studies explicitly mentioning the recurrence rates for Simpson grades I and II and Simpson grades III, IV, and V. These were analyzed to compare the recurrence rates, prevalence, and Odds Ratio between the two cohorts. The second group, a subset of the first one, included studies explicitly mentioning the recurrence rates for Simpson grade I and Simpson grade II to compare the recurrence rates, prevalence, and Odds Ratio between these two cohorts of patients. The proportions of recurrences among each Simpson grade were calculated for each study (prevalence per study). Odds Ratio (OR) values assessed categorical variables with 95% confidence intervals (CIs) and *p*-values. A restricted maximum likelihood (REML) “Fixed effect model” tests were adopted for meta-analysis. Cochran Q and I^2^ tests were adopted to determine whether the population under study deviated significantly from the general prevalence. An I^2^ value of less than 40% was defined as homogeneous. Peter’s test was used to detect possible publication bias. Meta-analyses were conducted for each of the two groups. All statistical tests were two-tailed. All statistical analyses were performed using GraphPad Prism (Insight Partners, New York, NY, USA; Version 10.1.2).

## 3. Results

### 3.1. Demographic and Clinical Features of the Institutional Cohort

According to the inclusion criteria, 74 consecutive patients were enrolled in the study. Among them, 63 (85.1%) were women, and 11 (14.9%) were men, with a median age of 61.92 (±13.94; range 24–84). The mean time to treatment (interval between the clinical onset and the operative procedure) was 5.21 years (±3.72; 0.12–12). The main presenting signs and symptoms were pyramidal (*n* = 30/74, 40%), followed by thoracic radicular pain (*n* = 27/74, 36%), ataxia or gait impairment (*n* = 6/74, 8%), hyposthenia (*n* = 5/74, 6%); six lesions were incidentally found on MRI scans for low back pain. The mean operative time was 293 min (±69.58), the mean ASA was 2.31 (±0.58), mean blood loss was 1.32 g/dL Hb points (±0.95). Tumors were most located in the thoracic spine (*n* = 60/74, 80%), followed by cervical location (*n* = 14, 20%). The dural tumoral base of attachment was anterior in 33 patients (45%), lateral in 22 (30%), and posterior in 19 (25%). The histopathological diagnoses were distributed as follows: 38 were psammomatous lesions (51%), 31 were transitional (42%), 3 were microcystic (4%), 1 was atypical (1%), 1 syncytial (1%), and 1 fibroblastic (1%). Three illustrative cases are exposed in Figure 1.

There was only one case of WHO grade II tumor, whereas the remaining 73 lesions were WHO grade I. The Simpson grade was so distributed: 16 patients received Simpson grade I, 43 grade II, 12 grade III, and 3 Simpson grade IV. Recurrences were observed in patients who underwent Simpson grade II and III resections: the first occurred four years after Simpson grade II, and another one two years after Simpson grade III. The mean follow-up was 92.43 months (±56.68; 1–15 years). Sample baseline characteristics are listed in Table 2.

### 3.2. Systematic Review

Figure 2 shows the PRISMA flowchart for the study selection process at different stages.

A total of 304 records were identified. Among them, 245 were found by searching electronic biomedical databases, and 59 studies were manually retrieved and cross-referenced with the corresponding reference lists of identified articles. English-language full text was unavailable for 32 records; hence, these were excluded. After removing 64 duplicates, 186 abstracts were screened for pertinence and methodology. Finally, 25 full-text articles were included in the systematic review and meta-analysis: 10 were case series, and 9 were retrospective cohort studies. The included studies and our institutional case series included 2142 individual patients operated for spinal meningioma.

### 3.3. Meta-Analysis

The study characteristics are presented in Table 5. Twenty-five articles were selected as suitable for the odds, ORs, and prevalence analyses. A total of 2142 spinal meningioma patients from 1980 to 2023 were included. Regarding the variables, 23 studies (94%) mentioned the respective recurrence rates for Simpson resection grade I and II and Simpson resection grade III, IV, and V, and 10 studies explicitly mentioned the recurrence rates for Simpson I and Simpson II. When comparing the studies, substantial homoscedasticity was shown between the enrolled samples (Cochran Q 20.59; df = 18; I^2^ = 17.47%). Therefore, a fixed-effect model was chosen for the meta-analysis. In the analysis of all the patients, the proportion of female patients (*n* 1718; 80.2%) was hugely higher than males (*n* 424; 19.7%) (Cochran Q 73.74; df = 18; I^2^ = 76.9); the mean age was 59.93 years (SD 14.99) (Cochran Q 23.60; df = 18; I^2^ = 27.96); the thoracic segment mainly was involved (*n* 1548; 72.3%), followed by the cervical segment (*n* 418; 19.5%), and lumbar (*n* 176; 7.9%) (Cochran Q 45.74; df = 18; I^2^ = 62.83%); the mean postoperative follow-up was 64.57 months (SD 31.19) (Cochran Q 75.75; df = 18; I^2^ = 72.29%).

Based on the fixed-effects model, the meta-analysis revealed a considerably greater recurrence prevalence in Simpson III, IV, and V compared with patients who received Simpson grades I and II (OR 0.10; CI95 0.06–0.16). Figure 3 illustrates the results of this analysis.

A further analysis of the subset of studies separately mentioning recurrences for Simpson grades I and II showed a significant prevalence of recurrence in Simpson II-treated spinal meningiomas compared to Simpson grade I procedures (OR 0.42; CI95 0.20–0.90). Figure 4 illustrates the results of this analysis.

### 3.4. Sensibility Analysis

A leave-one-out sensitivity analysis was conducted to evaluate the stability and robustness of the meta-analysis findings. By systematically removing each study from the dataset, the pooled odds ratios (ORs) were recalculated to identify any single study exerting a disproportionate influence on the overall results. The analysis demonstrated consistent pooled ORs across iterations, with minimal variation observed when individual studies were excluded. This stability in OR values suggests that the conclusions drawn from the meta-analysis are robust and not driven by any single study. Furthermore, the lack of significant fluctuation in the I-squared values across leave-one-out iterations indicates that heterogeneity was not excessively influenced by individual studies. (Table 6 and Table 7).

### 3.5. Publication Bias

Publication bias was evaluated using Peter’s test, which examines the relationship between effect sizes and their standard errors. The test yielded a coefficient of −0.2311 for the standard error (SE-logOR) with a *p*-value of 0.799. Given that this *p*-value is substantially higher than the conventional threshold of 0.05, there is no significant indication of publication bias in the meta-analysis. This result suggests that the effect sizes observed are likely representative and not skewed by selective publication.

## 4. Discussion

While meningiomas are the most common primary intracranial tumors of the central nervous system, spinal meningiomas are the most common primary spinal tumors [39].

Nevertheless, while many different demographic, clinical, neuroradiological, pathological, and therapeutic risk factors of recurrence have extensively been investigated for intracranial meningiomas [2], fewer data are available on meningiomas located in the spine. In this setting, factors associated with higher recurrence rates for spinal meningiomas are male sex [1,23], younger age [1], extradural extension in young patients [4], arachnoid scarring [27], ventral implant [27], Ki67 and arachnoid invasion [40], and higher WHO grade [30].

Regarding the role of the extent of resection according to Simpson grading as a predictor of recurrence in spinal meningiomas, there is no unanimous consent in the pertinent literature with contrasting results, with some studies reporting no difference [1,40,41] and others reporting a higher recurrence rate associated with Simpson grade II versus grade I [1,37]. 

The results of the present meta-analysis revealed a considerably higher recurrence rate in Simpson grades III, IV, and V compared with patients who received Simpson grades I and II (OR 9.789; CI95 6.087–15.743) (Figure 3). Furthermore, a significant prevalence of recurrence in Simpson II-treated spinal meningiomas compared to Simpson grade I procedures (OR 2.271; CI95 1.018–5.068).

Functional outcomes following surgical resection of spinal meningiomas were seldom reported in the analyzed literature, limiting comprehensive comparisons. Among the studies reviewed, Simpson grades I and II consistently demonstrated superior functional outcomes compared to grades III, IV, and V [28]. Patients undergoing Simpson grade I resections exhibited the most favorable recovery, with minimal long-term deficits [6]. While Simpson grade II resections provided good postoperative recovery, outcomes were slightly less favorable [6]. Grades III and IV, characterized by subtotal resection, were associated with poorer functional outcomes and persistent neurological deficits [8].

Surgery represents the first line of treatment, with the goal of a gross total (Simpson grade I—macroscopically complete removal of the lesion including its dural attachment and any abnormal bone) and en-bloc tumor resection while preserving/restoring or arresting the deterioration of the neurological functions and without harming the spinal instability, to avoid or decrease the rate of recurrence [42,43]. Nevertheless, this purpose is not always achievable due to the anatomical relationship of the tumor with a high functional structure like the spinal cord; therefore, a Simpson grade II (macroscopically complete removal of the lesion and coagulation of its dural attachment) is an acceptable alternative. Particularly, tumors with ventral attachment represent a technical challenge for Simpson grade I. In addition, the risk of postoperative complications associated with Simpson grade I, including neurological complications, pseudomeningoceles, and CSF leak, should be considered. In this scenario, Saito et al. [44]. developed an alternative technique in which after tumor exeresis, only the inner dural layer is resected, preserving the outer dural layer, to avoid the technical difficulties in the dural reconstruction and the postoperative complications associated.

Therefore, even if the resection of the pathological dura (Simpson I) is associated with a lower risk of recurrence compared to its coagulation and preservation (Simpson II), the strategy of treatment should be carefully balanced between pros and cons and several features correlated to the pathology and the pathology should be taken into account [39]. Remembering the mantra “*Primum non nocere*”, for ventral spinal meningiomas in elderly symptomatic patients with short life expectancy, the tumor debulking associated with dural coagulation is advisable.

In spinal meningiomas, achieving Simpson grade I resection is often limited by anatomical constraints and the risk of postoperative instability, particularly in ventral lesions. As a result, Simpson grade II resection frequently represents the most viable balance between safety and recurrence risk.

In symptomatic young or adult patients with ventral tumoral implants, a Simpson grade II aiming clinical symptoms and signs resolution, with the option of a re-operation if symptoms will occur over the years, represents a valid option, mainly when intraoperative biopsy detects a WHO grade I tumor.

While Simpson grading provides a valuable framework, WHO grading remains a critical factor influencing outcomes. Grade I lesions showed excellent prognosis, but grades II and III were associated with significantly higher recurrence rates, highlighting the need for tailored surgical and postoperative strategies.

The surgical procedure is tailored case by case according to the patient and pathology features, and involves a midline posterior approach to the spinal canal, through laminectomy or hemilaminectomy, followed by a midline linear dural incision and tumor resection, respecting the spinal cord or exerting gentle traction to gain the tumor access [37]. Intraoperative neurophysiological monitoring is mandatory in this surgery since it changes the functional postoperative outcome dramatically [38]. Techniques for meningioma resection vary depending on the dural attachment of the meningioma and the consistency and extent of the tumor [31]. Intuitively, the extent of resection of the meningioma depends mainly on the dural attachment [7]. For posterior meningiomas, a Simpson I is more likely to be achieved; for ventrally attached lesions, removing the involved dura mater is challenging [45]. Conversely, the procedure is also challenging for lesions extruding laterally into the conjugation foramina. The calcific consistency is a further issue [44]. However, in the literature, spinal meningiomas with posterior dural attachment are described as more easily resectable along with their dural implant, configuring a Simpson I grade [46]. On the other hand, for ventral meningiomas, a Simpson grade II resection has been proposed as acceptable [5]. In the literature, the recurrence rate for microsurgically operated spinal meningiomas varies between 1.3 and 14.7% [21], with disparities due to gender [27], age [8], dural attachment [33], WHO grade [26], and degree of surgical resection [32].

This study demonstrated the standard of care for spinal meningiomas should be Simpson grade I, where achievable, since this was statistically correlated with a significant decrease in recurrence risk. This study represents the first significant evidence of the validity of Simpson grading for spinal meningiomas, after more than 50 years since its adoption for meningioma surgery.

A limitation of this study was that it did not consider further variables for analysis, such as dural attachment, histology, WHO grade, and patient age, in multivariate analysis. Regrettably, no studies describing recurrence rates are available for these variables among studies enrolled in this meta-analysis.

The paucity of data about functional recovery across the included studies represents a significant limitation. Most literature prioritized recurrence and tumor control, underscoring the need for future studies to systematically document neurological outcomes and quality of life to better guide surgical management.

Tumor location and extension are critical factors influencing Simpson grade and surgical outcomes. While these parameters were included in our institutional cohort, their inconsistent reporting across the included studies limited their integration into the meta-analysis, representing a notable limitation.

Another limitation is the lack of data on the timing of relapses for the different Simpson grades. Thereafter, a limitation concerns the low number of studies reporting Simpson grade I and II separately, given the common association with gross total resection of spinal meningioma. A large-scale prospective study should be performed to provide robust evidence about the predictive factors for recurrence in spinal meningiomas.

A large-scale prospective study should be performed to provide robust evidence about the predictive factors for recurrence in spinal meningiomas.

This study represents strong evidence about the prognostic role of the Simpson grading system in spinal meningiomas. Whenever possible, the dural margins and overlying bone should be resected together with the spinal meningioma in surgery to minimize the recurrence rates of spinal meningiomas in the long term.

## 5. Conclusions

This systematic review and meta-analysis revealed the validity of Simpson’s grading to predict the recurrence rate according to the extent of resection in spinal meningiomas; therefore, it represents useful information for driving the decision-making process of management strategy. Surgery is the first line of treatment with the goal of maximal safe tumor resection (Simpson grade I) while preserving/restoring or arresting deterioration of neurological functions; nevertheless, when the risk of postoperative complications is high, especially for tumors with ventral tumoral dural attachment, the Simpson grade II aiming clinical resolution is a valid alternative, with the option of multiple reoperations over the years if symptoms will occur. The results of this study suggest that the standard of care for spinal meningiomas should be reconsidered, given the significant predictive value of the Simpson grading system for spinal meningiomas. 

## Figures and Tables

**Figure 1 cancers-17-00034-f001:**
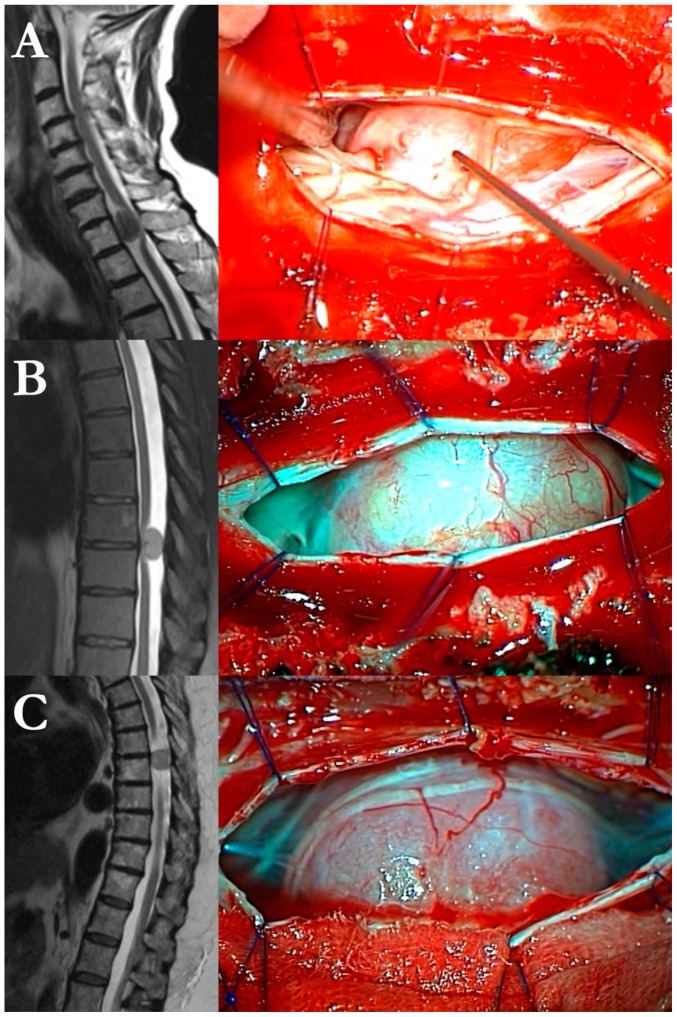
Three illustrative cases from our series of 74 spinal meningiomas: left, preoperative T1-weighted MRI scans; right, intraoperative images. (**A**) T1 transitional meningioma with anterior growth. (**B**) T7–T8 psammomatous/calcific meningioma with posterolateral extension. (**C**) T7 fibroblastic meningioma with right lateral extension.

**Figure 2 cancers-17-00034-f002:**
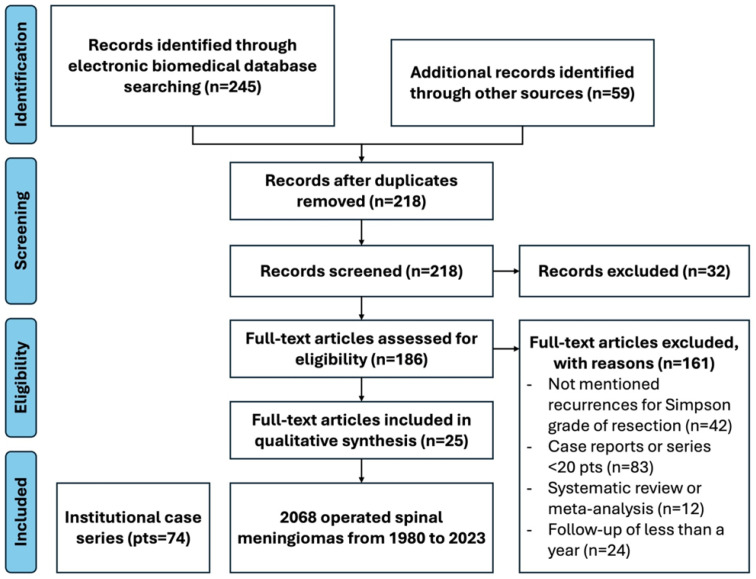
PRISMA flowchart for study selection.

**Figure 3 cancers-17-00034-f003:**
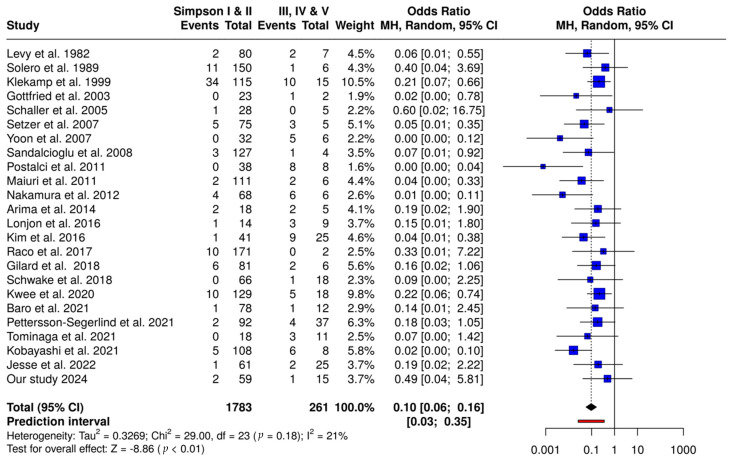
Forest plot comparing Simpson I vs. Simpson II resections. This plot presents a meta-analysis of recurrence rates for patients with Simpson grade I versus Simpson grade II resections. Each study’s odds ratio (OR) and 95% confidence interval (CI) are shown, with the overall OR calculated using a Mantel–Haenszel random-effects model. The pooled OR is 0.42 (95% CI: 0.20–0.90), indicating a statistically significant advantage for Simpson grade I resections in reducing recurrence rates (*p* < 0.05). The heterogeneity assessment shows no significant variability among studies (I^2^ = 0%), supporting the consistency of effect sizes across studies [1,3,6,8,12,19,20,22,24,25,26,27,28,29,30,31,32,33,34,35,36,37,38].

**Figure 4 cancers-17-00034-f004:**
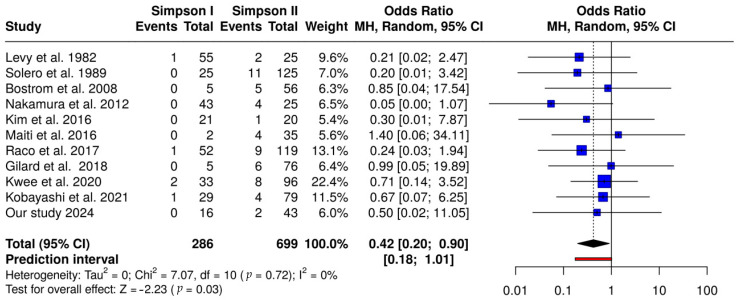
Forest plot comparing Simpson I and II vs. Simpson III, IV, and V resections. This plot illustrates a meta-analysis comparing recurrence rates for combined Simpson grades I and II versus Simpson grades III, IV, and V resections. Individual study ORs and 95% CIs are provided, with an overall pooled OR of 0.10 (95% CI: 0.06–0.16), suggesting a substantial reduction in recurrence risk for grades I and II compared to grades III, IV, and V (*p* < 0.01). Significant heterogeneity was detected (I^2^ = 21%), indicating variability in effect sizes across studies [1,6,19,20,21,22,23,24,25,26].

**Table 1 cancers-17-00034-t001:** Simpson grading system on meningioma resection.

Simpson Grade	Definition
I	Macroscopically complete tumor resection with removal of affected dura and underlying bone
II	Macroscopically complete tumor resection with coagulation of affected dura only
III	Macroscopically complete tumor resection without removal of affected dura or underlying bone
IV	Subtotal tumor resection
V	Decompression with or without biopsy

**Table 2 cancers-17-00034-t002:** Institutional case series consisting of 74 consecutive spinal meningiomas, from May 2006 to January 2023.

No. of Patients *	74 (100)
Sex *	
Male	11 (15)
Female	63 (85)
Age † (years)	61.92 (±13.94; 24–84)
Segment *	
Cervical (C2–C7)	14 (19)
High thoracic (T1–T4)	12 (16)
Middle thoracic (T5–T8)	14 (19)
Lower thoracic (T9–T12)	34 (46)
Lumbosacral (L1-S1)	0
Dural implant *	
Anterior	33 (45)
Lateral	22 (30)
Posterior	19 (25)
Symptom duration before surgery † (Years)	5.21 (±3.72; 0.12–12)
Exordium signs and symptoms *	
Pyramidal signs	30 (40)
Radicular band pain	27 (36)
Ataxia or gait impairment	6 (8)
Hyposthenia	5 (6)
Incidentally found	6 (8)
Operative time † (mins)	293 min (±69.58)
Intraoperative blood loss † (Hb g/dL)	1.32 (±0.95)
Postoperative degency † (days)	293 (±69.58)
Histopathology *	
Psammomatous	38 (51)
Transitional	31 (42)
Microcystic	3 (4)
Syncytial	1 (1)
Fibroblastic	1 (1)
Atypical	1 (1)
ASA †	2.31 (±0.58)
Calcific lesions *	21 (28)
Simpson resection grade *	
I	16 (22)
II	43 (58)
III	12 (16)
IV	3 (4)
V	0
Recurrencies *	2 (3)
Follow-up † (months)	92.43 (±56.68; 1–15 years)

* Categorical variables are expressed as raw frequencies (%). † Qualitative variables are expressed as mean (±SD); range.

**Table 3 cancers-17-00034-t003:** The search strategy of the network meta-analysis. Data regarding the histology, the involved spinal level, the clinical and anthropometric parameters of the patients, the Simpson grade of resection, and the recurrence rates were collected from full texts when available. G.C. and S.C. performed data extraction.

Frame	Mesh Terms	Search	Inclusion Criteria	Exclusion Criteria	Sources
P (patients, participants, population)	#1 “Spinal meningioma” #2 “Adult spinal surgery” OR “Intradural extramedullary tumor”	#1 AND #2 AND #3 AND #4 AND #5	Published in peer-review journals	Irrelevant title or abstract Irrelevant full text Editorial, reviews, meta-analysis Studies with less than 20 subjects Experimental/nonhuman studies Mean follow up less than one year	Databases (PubMed, Cochrane Library, ClinicalTrials.gov, Web of Science, and Scopus)
I (intervention)	#3 “Laminectomy”		English language		
C (comparator)	#4 “Simpson I” OR “Simpson II” OR “Simpson III, IV, V”		Randomized controlled trials, non-randomized observational studies, case series		
O (outcome)	#5 “Recurrence” OR “Reoperation” OR “Relapsed spinal meningioma”		Accurately described sample characteristics, Simpson resection grade, stratified recurrences, histopathological diagnosis		Reference list
T (time)	The search period: 1980 until December 2023				

**Table 4 cancers-17-00034-t004:** Criteria and definitions used in the current meta-analysis for grouping of complications reported among the eligible studies.

Event(s)	Definition
Recurrence	Tumor relapse for Simpson grades I and II, and lesion volume doubled for Simpson III and IV, requiring reoperation.
Reintervention	Surgical intervention on the operated patient, due to a relapse of the tumor, on the same level as the previous surgery.
No recurrence	No visible lesion at the operated level, at the last follow-up.
Simpson grading scale	Postoperative surgical scale to assess the extent of surgical resection in meningioma. Grade I: Complete removal, including resection of the underlying bone and associated dura. Grade II: Complete removal with coagulation of dural attachment. Grade III: Complete removal without resection of dura or coagulation. Grade IV: Subtotal resection. Grade V: Simple decompression with or without biopsy.

**Table 5 cancers-17-00034-t005:** General characteristics of eligible studies. * The value is expressed in months. † Level of evidence according to Melnyk & Fineout-Overholt 2023. ¥ For studies not separately mentioning Simpson grades I and II.

Author	Year	Country	Study Design	Recruitment Period		Patients		Simpson I		Simpson II		Simpson III, IV, V		Mean Follow Up *	Max Follow Up *	Level of Evidence †
				From	To	Sample	Sex (F-M)	Pts	Ev	Pts	Ev	Pts	Ev			
Levy et al. [19]	1982	USA	CS	1946	1982	97	78-19	55	0	25	2	7	2	84	336	6
Solero et al. [20]	1989	Italy	CS	1954	1983	174	143-31	25	0	125	11	6	1	48	384	6
Bostrom et al. [21]	2008	Germany	R/Coh	1990	2005	61	50-11	5	0	56	5	-	-	31.3	120	4
Nakamura et al. [6]	2012	Japan	CS	1983	2006	68	56-12	43	0	25	4	6	6	145.2	264	6
Kim et al. [22]	2016	South Korea	CS	1989	2003	55	40-15	21	0	20	1	25	9	45	156	6
Maiti et al. [23]	2016	USA	CS	2001	2015	38	31-7	2	0	35	4	-	-	51.2	82	6
Raco et al. [24]	2017	Italy	R/Coh	1976	2011	173	138-35	52	1	119	9	2	0	50.8	120	4
Gilard et al. [25]	2018	France	CS	2009	2016	87	70-17	5	0	76	6	6	2	92.4	300	6
Kwee et al. [26]	2020	Netherlands	R/Coh	1989	2018	166	139-27	33	2	96	8	18	5	12	276	4
Kobayashi et al. [1]	2021	Japan	R/Coh	1998	2018	116	94-22	29	1	79	4	8	6	84.8	252	4
Our study	2024	Italy	R/Coh	2006	2023	74	63-11	16	0	43	2	15	1	92.4	180	4
										Simpson I, II ¥		Simpson III, IV, V				
Klekamp et al. [27]	1999	Germany	R/Coh	1977	1998	130	93-24	-	-	115	34	15	10	20	156	4
Gottfried et al. [28]	2003	USA	CS	1992	2002	25	19-6	-	-	23	0	2	1	23	64	6
Schaller et al. [29]	2005	Germany	R/Coh	1980	1995	33	30-3	-	-	28	1	5	0	96	180	4
Setzer et al. [30]	2007	USA	R/Coh	1999	2007	80	58-22	-	-	75	5	5	3	43.5	142.7	4
Yoon et al. [31]	2007	South Korea	CS	1970	2005	38	31-7	-	-	32	0	6	5	73	223	6
Sandalcioglu et al. [32]	2008	Germany	CS	1990	2006	131	114-17	-	-	127	3	4	1	61	116	6
Postalci et al. [33]	2011	Turkey	CS	1995	2009	46	33-13	-	-	38	0	8	8	60	60	6
Maiuri et al. [11]	2011	Italy	R/Coh	1986	2008	117	87-30	-	-	111	2	6	2	144	276	4
Arima et al. [3]	2014	Japan	CS	2007	2014	23	15-8	-	-	18	2	5	2	32.1	84	6
Lonjon et al. [34]	2016	UK	CS	2004	2014	22	16-7	-	-	14	1	9	3	40	146	6
Schwake et al. [35]	2018	Germany	CS	1996	2016	84	53-31	-	-	66	0	18	1	19	40	6
Baro et al. [36]	2021	Italy	R/Coh	2011	2018	90	75-15	-	-	78	1	12	1	17	75	4
Pettersson-Segerlind et al. [8]	2021	Sweden-Denmark	R/Coh	2005	2017	129	106-23	-	-	92	2	37	4	98.4	192	4
Tominaga et al. [37]	2021	Japan	CS	1992	2010	29	22-7	-	-	18	0	11	3	132	160.5	6
Jesse et al. [38]	2022	Switzerland	CS	2009	2021	86	75-11	-	-	61	1	25	2	29.8	162.6	6

**Table 6 cancers-17-00034-t006:** Leave-one-out sensitivity analysis for Simpson I and II vs. Simpson III, IV, V. This table summarizes the leave-one-out sensitivity analysis for the comparison between Simpson I and II versus Simpson III, IV, and V resections. Each row displays the pooled OR and 95% CI after systematically excluding one study. The results show consistent pooled ORs around 6.415, indicating the robustness of the meta-analysis findings. Heterogeneity remained low (I^2^ < 40%) throughout, underscoring the consistency of effect sizes across the included studies.

Excluded_Study	Pooled_Log_Or	Pooled_Or	Ci_Low	Ci_High	Pooled_Se	I_Squared
Levy et al. [19]	1.859	6.415	4.067	10.119	0.233	<40%
Solero et al. [20]	1.905	6.721	4.254	10.619	0.233	<40%
Klekamp et al. [27]	1.858	6.409	3.943	10.417	0.248	<40%
Gottfried et al. [28]	1.999	7.378	4.598	11.84	0.241	<40%
Schaller et al. [29]	1.874	6.516	4.141	10.253	0.231	<40%
Yoon et al. [31]	1.823	6.19	3.921	9.771	0.233	<40%
Setzer et al. [30]	1.905	6.721	4.254	10.619	0.233	<40%
Sandalcioglu et al. [32]	1.851	6.366	4.037	10.037	0.232	<40%
Postalci et al. [33]	1.797	6.032	3.824	9.516	0.233	<40%
Maiuri et al. [11]	1.8	6.047	3.81	9.599	0.236	<40%
Nakamura et al. [6]	1.776	5.907	3.735	9.341	0.234	<40%
Arima et al. [3]	1.883	6.572	4.162	10.377	0.233	<40%
Raco et al. [24]	1.861	6.432	4.055	10.203	0.235	<40%
Kwee et al. [26]	1.979	7.235	4.46	11.738	0.247	<40%
Pettersson-Segerlind et al. [8]	1.888	6.604	4.153	10.5	0.237	<40%
Tominaga et al. [37]	1.905	6.721	4.262	10.599	0.232	<40%
Kobayashi et al. [1]	1.909	6.743	4.29	10.601	0.231	<40%
Jesse et al. [38]	1.883	6.571	4.165	10.367	0.233	<40%
Our study	1.887	6.597	4.19	10.386	0.232	<40%

**Table 7 cancers-17-00034-t007:** Leave-one-out sensitivity analysis for Simpson I vs. Simpson II. This table presents the results of a leave-one-out sensitivity analysis, showing the pooled odds ratio (OR) and 95% confidence intervals (CI) after excluding each study from the Simpson I vs. Simpson II meta-analysis. The stability of the pooled OR, which remains around 2.169, suggests that no single study disproportionately influences the overall results. Heterogeneity was consistently low, with I^2^ < 40%, indicating uniformity in effect sizes across studies.

Excluded_Study	Pooled_Log_Or	Pooled_Or	Ci_Low	Ci_High	Pooled_Se	I_Squared
Levy et al. [19]	0.774	2.169	0.938	5.016	0.428	<40%
Solero et al. [20]	0.687	1.988	0.833	4.747	0.444	<40%
Bostrom et al. [21]	0.81	2.247	0.953	5.299	0.438	<40%
Nakamura et al. [6]	0.459	1.582	0.666	3.757	0.441	<40%
Raco et al. [24]	0.898	2.454	1.041	5.786	0.438	<40%
Kwee et al. [26]	1.114	3.047	1.22	7.609	0.467	<40%
Kobayashi et al. [1]	0.982	2.67	1.155	6.17	0.427	<40%
Our study	0.872	2.391	1.033	5.534	0.428	<40%

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
