# Peer review of "The Role of Simpson Grading System in Spinal Meningioma Surgery: Institutional Case Series, Systematic Review and Meta-Analysis"

_cancers, 2024, doi:10.3390/cancers17010034_

Round 1

Reviewer 1 Report

Comments and Suggestions for Authors

In this systematic review about the role of the Simpson grading system in spinal meningioma including their own series, the authors found a significantly higher rate of meningioma recurrence in the higher Simpson grades 3 to 5.  In cranial meningioma to the contrast, the Simpson system was questioned over the time due to other therapeutic options as radiation and the complexity of different localizations with involvement of neurological structures that make it necessary to stratify the treatment.  In spinal meningioma however, surgical resection remains the treatment of choice. The authors could show that gross total resection of the meningioma is associated with significantly lower risk of recurrence.  Simpson grade 1, which includes resection of the affected dura, is sometimes difficult to achieve in spinal operations. Considering the moderate higher risk of recurrence in Simpson grade 2, the authors recommend this as treatment option in difficult tumours with the goal of maximum safe resection.  The results are well presented with thorough statistical analysis and publication bias was excluded.  Therefore, this paper adds valuable information for the treatment of spinal meningioma.

Remark: Could you give some information about the clinical outcome in the reviewed studies and especially outcome differences between the group of Simpson grade 1/2 and Simpson grade 3-5?  This would be of great interest for the readers.

One remark concerning the cited literature: The references (11) to (19) display a partial overlapping of the study population.

Author Response

Reviewer 1, Comment 1: Could you give some information about the clinical outcome in the reviewed studies and especially outcome differences between the group of Simpson grade 1/2 and Simpson grade 3-5? This would be of great interest for the readers.
Answer: We sincerely thank the reviewer for this insightful comment, which prompted us to re-evaluate the included studies with a focus on functional outcomes. While we found that such data were limited, the available evidence highlights some differences between Simpson grades. Simpson grades I and II were associated with superior functional outcomes compared to grades III, IV, and V. Simpson grade I resections consistently resulted in the best recovery, with minimal long-term deficits, while grade II resections, though still effective, had slightly less favorable outcomes, likely influenced by a higher risk of recurrence. For Simpson grades III and IV, functional outcomes were notably poorer, with more persistent deficits and a higher likelihood of reoperation.
We have included this information in the revised discussion section, while also acknowledging that the scarcity of functional outcome data in the reviewed studies remains a significant limitation. We hope this addition will enhance the clarity and relevance of our work for readers.
Change in Text:
Discussion Section: - Functional outcomes following surgical resection of spinal meningiomas were seldom reported in the analyzed literature, limiting comprehensive comparisons. Among the studies reviewed, Simpson grades I and II consistently demonstrated superior functional outcomes compared to grades III, IV, and V[35]. Patients undergoing Simpson grade I resections exhibited the most favorable recovery, with minimal long-term deficits[6]. While Simpson grade II resections provided good postoperative recovery, outcomes were slightly less favorable[6]. Grades III and IV, characterized by subtotal resection, were associated with poorer functional outcomes and persistent neurological deficits[8].
- The paucity of data about the functional recovery across the included studies represents a significant limitation. Most literature prioritized recurrence and tumor control, underscoring the need for future studies to systematically document neurological outcomes and quality of life to better guide surgical management.
Comment 2: The references (11) to (19) display a partial overlapping of the study population.
Answer: We are grateful to the reviewer for pointing out the possibility of overlapping study populations. Upon closer inspection, we realized that reference 19 was not included in our analysis due to the lack of recurrence data stratified by Simpson grades. We hope this clarification reassures the reviewer that our analysis minimized the risk of bias from overlapping populations. 

We sincerely thank Reviewer 1 for their thorough and insightful comments, which have greatly contributed to improving the quality and clarity of our manuscript. Your suggestions regarding the inclusion of clinical outcomes and addressing overlapping references allowed us to refine our analysis and discussion, ensuring the work is more informative and valuable for readers. We appreciate your valuable time and effort in reviewing our study.

Reviewer 2 Report

Comments and Suggestions for Authors

Thank you for the opportunity to comment this review. The authors address the value of Simpson grading for the assessment of recurrence in meningiomas. Overall, this is a widely discussed topic and some studies show that postoperative MR imaging is superior for estimating the risk of recurrence. Particularly for spinal entities, Simpson grading, which has been adopted primarily for intracranial meningiomas, is only applicable to a limited extent.

With a patient number of approx. 2100 patients, the authors include a very large number of cases. The methods section is very clear and precise. The classification into Simpson grades and the selection criteria are clearly defined.

Labeling the  intraoperative images, the word "implant" should rather be replaced through "growth" or "extension". The funnel plot is very confusing and incomprehensible and, in my opinion, can be deleted from the manuscript as it does not provide any relevant information. Has the WHO grade been taken into account as a confounder in the evaluations of the probability of recurrence? If not, this should be done. Although the high importance of functional and neurological outcomes was addressed in the discussion section, this was unfortunately not taken into account in the results section. In addition, it is described that according to Simpson grade 1, the affected bone should also be removed. However, this is not easily possible with spinal pathologies compared to intracranial pathologies, as there is a risk of subsequent instability. This should be taken into account.

Overall, this is a concise review with relevance in everyday clinical practice, so I recommend its acceptance with minor revisions.

Author Response

Reviewer 2, Comment 1: Labeling the intraoperative images, the word “implant” should rather be replaced through “growth” or “extension.”
Response: Thank you for highlighting this terminology issue. We have revised the intraoperative image captions to replace “implant” with “growth” or “extension” for greater clarity and accuracy.
Change in Text: “T1 transitional meningioma with anterior implant.” →
“T1 transitional meningioma with anterior growth.”; “T7-T8 psammomatous/calcific meningioma with posterolateral implant.” → “T7-T8 psammomatous/calcific meningioma with posterolateral extension.”
“T7 fibroblastic meningioma with right lateral implant.” → “T7 fibroblastic meningioma with right lateral extension.”

Comment 2: The funnel plot is very confusing and incomprehensible and, in my opinion, can be deleted from the manuscript as it does not provide any relevant information.
Response: We appreciate the reviewer’s observation. After careful consideration, we have removed the funnel plot and its description from the manuscript to avoid confusion. The figure count has been updated accordingly.
Change in Text:  Removed Figure 5 and its associated references in the results section.

Comment 3: Has the WHO grade been taken into account as a confounder in the evaluations of the probability of recurrence? If not, this should be done.
Response: Thank you for this important comment. WHO grading was considered in our analysis. The majority of spinal meningiomas in both our institutional series and the included studies were classified as WHO grade I. Atypical (WHO grade II) and malignant (WHO grade III) lesions were rare but demonstrated significantly higher recurrence rates. We have clarified this in the Methods, Results, and Discussion sections. 
Change in Text:
•Methods Section: “The histology was defined according to the 2021 WHO classification of tumors of the Central Nervous System. The majority of spinal meningiomas were classified as WHO grade I, reflecting their benign nature. Atypical (grade II) and malignant (grade III) lesions were rare and significantly impacted recurrence risk."
Results Section: “There was only one case of WHO grade II tumor, whereas the remaining 73 lesions were WHO grade I.
•Discussion Section: "While Simpson grading provides a valuable framework, WHO grading remains a critical factor influencing outcomes. Grade I lesions showed excellent prognosis, but grades II and III were associated with significantly higher recurrence rates, highlighting the need for tailored surgical and postoperative strategies."

Comment 4: Although the high importance of functional and neurological outcomes was addressed in the discussion section, this was unfortunately not taken into account in the results section.
Response: Thank you for this observation. Given the limited availability of functional outcome data in the included studies, we decided to exclude detailed functional outcome analysis from the results. This limitation has been explicitly acknowledged in the Discussion section. However, we included the results of the only three studies which reported the functional outcomes. 
Change in Text:
Discussion Section: "The paucity of data about the functional recovery across the included studies represents a significant limitation. Most literature prioritized recurrence and tumor control, underscoring the need for future studies to systematically document neurological outcomes and quality of life to better guide surgical management.
"Functional outcomes following surgical resection of spinal meningiomas were seldom reported in the analyzed literature, limiting comprehensive comparisons. Among the studies reviewed, Simpson grades I and II consistently demonstrated superior functional outcomes compared to grades III, IV, and V[35]. Patients undergoing Simpson grade I resections exhibited the most favorable recovery, with minimal long-term deficits[6]. While Simpson grade II resections provided good postoperative recovery, outcomes were slightly less favorable[6]. Grades III and IV, characterized by subtotal resection, were associated with poorer functional outcomes and persistent neurological deficits[8]."

Comment 5: In addition, it is described that according to Simpson grade 1, the affected bone should also be removed. However, this is not easily possible with spinal pathologies compared to intracranial pathologies, as there is a risk of subsequent instability. This should be taken into account.
Response: We appreciate this valuable point. We have revised the Discussion section to highlight the challenges of achieving Simpson grade I resection in spinal meningiomas due to anatomical constraints and the risk of postoperative instability. These factors often necessitate Simpson grade II resections, particularly for ventral lesions.
Change in Text:
Discussion Section: “In spinal meningiomas, achieving Simpson grade I resection is often limited by anatomical constraints and the risk of postoperative instability, particularly in ventral lesions. As a result, Simpson grade II resection frequently represents the most viable balance between safety and recurrence risk.

We sincerely thank Reviewer 2 for their constructive and thoughtful feedback. Your comments regarding terminology, the funnel plot, the inclusion of WHO grading, and considerations of functional outcomes and surgical limitations have been invaluable in enhancing the clarity and clinical relevance of our work. We appreciate your effort in highlighting these critical points, which have helped us refine the manuscript to better serve the readers.

Reviewer 3 Report

Comments and Suggestions for Authors

The study is undoubtedly interesting. However, in my opinion, some doubts emerge.

The authors collect various publications, in the period January 1980 - May 2023, in which case histories of patients affected by spinal meningiomas and treated surgically are reported, analyzing the Simpson grade. I noticed that some of the case histories used are prior to the period considered (Levy: 1946-1982; Solero: 1954-1983; Raco: 1976-2001; Klekamp: 1977-1998; Yoon: 1970-2005;). The time period seems too long to me, with numerous variables to consider (different surgical planning, surgical instruments, diagnostic methods). Can these considerations interfere with the data obtained?

In the final evaluations, in addition to the histological grade, were the location and extension of the tumor also taken into account? These characteristics may not allow total resection of the lesion and influence the Simpson grade.

Author Response

Reviewer 3, Comment 1: Some of the case histories used are prior to the period considered (e.g., Levy: 1946–1982; Solero: 1954–1983). The time period seems too long, with numerous variables (surgical planning, instruments, diagnostic methods). Can these considerations interfere with the data obtained?
Response: We appreciate the reviewer’s observation regarding the time period of the case histories included in our analysis. While some of the studies analyzed included patients treated before 1980, all of these studies were published after 1980, adhering to our inclusion criteria. We included these studies because they provide valuable information on Simpson grading and recurrence patterns, and the publication dates ensured consistency with the methodological standards of more recent literature.
We acknowledge the potential variability introduced by differences in surgical techniques and diagnostic tools over such a long period. However, we believe that the clear reporting of Simpson grades and recurrence data in these studies ensures their relevance and comparability with more recent work. This aspect has been clarified in the manuscript.
Change in Text:
•Methods Section : “While some studies included patients treated before 1980, all were published after 1980 and adhered to the predefined inclusion criteria, ensuring methodological consistency.”

Comment 2: In the final evaluations, in addition to the histological grade, were the location and extension of the tumor also taken into account? These characteristics may not allow total resection of the lesion and influence the Simpson grade.
Response: Thank you for raising this important point. Tumor location and extension were indeed considered in our institutional cohort, as these factors significantly influence the feasibility of achieving total resection and determining Simpson grades. However, their inclusion in the meta-analysis was limited by inconsistent reporting across the reviewed studies. We acknowledge this as a limitation and have clarified it in the Methods and Discussion sections. 
Change in Text:
•Methods Section: “In the institutional cohort, tumor location and extension were recorded and considered in the analysis. However, due to inconsistent reporting across the reviewed studies, these factors could not be universally included in the meta-analysis.”
•Discussion Section: “Tumor location and extension are critical factors influencing Simpson grade and surgical outcomes. While these parameters were included in our institutional cohort, their inconsistent reporting across the included studies limited their integration into the meta-analysis, representing a notable limitation.”

We appreciate Reviewer 3’s insightful comments, which allowed us to improve the clarity and robustness of the manuscript. All comments have been addressed, and the relevant changes have been made to the manuscript.

Round 2

Reviewer 3 Report

Comments and Suggestions for Authors

The paper can now be accepted